# Both Nuclear and Membrane Estrogen Receptor Alpha Impact the Expression of Estrogen Receptors and Plasticity Markers in the Mouse Hypothalamus and Hippocampus

**DOI:** 10.3390/biology12040632

**Published:** 2023-04-21

**Authors:** Sanoara Mazid, Elizabeth M. Waters, Chloe Lopez-Lee, Renata Poultan Kamakura, Batsheva R. Rubin, Ellis R. Levin, Bruce S. McEwen, Teresa A. Milner

**Affiliations:** 1Feil Family Brain and Mind Research Institute, Weill Cornell Medicine, 407 East 61st Street, New York, NY 10065, USA; 2Harold and Margaret Milliken Hatch Laboratory of Neuroendocrinology, The Rockefeller University, 1230 York Avenue, New York, NY 10065, USA; 3Molecular Biology and Biochemistry, University of California, Irvine, 3205 McGaugh Hall, Irvine, CA 92697-3900, USA

**Keywords:** neural plasticity, phosphorylated TrkB, estrogen receptor beta, mossy fiber pathway

## Abstract

**Simple Summary:**

The diversity of membrane and nuclear estrogen receptor (ER) expression in the brain suggests numerous roles for estrogens throughout life. The aim of this study was to examine the effects of selective expression of membrane-only ERα (MOER) or nuclear-only ERα (NOER) on the distributions of proteins known to be involved in paraventricular hypothalamic nucleus (PVN) function and estrogen-mediated plasticity in the hippocampus. We found that, although the PVN has both membrane and nuclear ERα, only loss of nuclear ERα impacted ERβ expression. In hippocampal CA1, an area with primarily extranuclear ERα, and the dentate gyrus, which contains nuclear and extranuclear ERα and ERβ, a marker of plasticity (phosphorylated Trkb) changed with both membrane and nuclear receptor loss, but the direction of the change was region-specific. In the hippocampal dentate gyrus and CA3 regions, where there is less ERα expression overall, loss of nuclear or membrane ERα appeared to have less impact on plasticity markers. Altogether, this work generates new questions about the functions of membrane and nuclear ERs in the development of the brain and actions in adulthood.

**Abstract:**

Estrogens via estrogen receptor alpha (ERα) genomic and nongenomic signaling can influence plasticity processes in numerous brain regions. Using mice that express nuclear only ERα (NOER) or membrane only ERα (MOER), this study examined the effect of receptor compartmentalization on the paraventricular nucleus of the hypothalamus (PVN) and the hippocampus. The absence of nuclear and membrane ERα expression impacted females but not males in these two brain areas. In the PVN, quantitative immunohistochemistry showed that the absence of nuclear ERα increased nuclear ERβ. Moreover, in the hippocampus CA1, immuno-electron microscopy revealed that the absence of either nuclear or membrane ERα decreased extranuclear ERα and pTrkB in synapses. In contrast, in the dentate gyrus, the absence of nuclear ERα increased pTrkB in synapses, whereas the absence of membrane ERα decreased pTrkB in axons. However, the absence of membrane only ERα decreased the sprouting of mossy fibers in CA3 as reflected by changes in zinc transporter immunolabeling. Altogether these findings support the idea that both membrane and nuclear ERα contribute overlapping and unique actions of estrogen that are tissue- and cellular-specific.

## 1. Introduction

Estrogens activate both nuclear and membrane estrogen receptor (ERs) in the brain [1]. Nuclear ERs modulate gene transcription, whereas membrane ERs act via signal transduction, which primarily impacts plasticity locally [2]. Additionally, there is a possibility of some crosstalk between membrane and nuclear steroid receptors in which membrane-associated signaling can potentiate nuclear transcription [3,4,5]. ERα and ERβ have overlapping abilities but distinct actions [6,7].

ERα is expressed throughout the whole brain, and each brain area has distinct patterns of nuclear and membrane ERα expression. Notably, the mouse paraventricular nucleus of the hypothalamus (PVN) has prominent nuclear ERα, with a lower amount of extranuclear ERα [8]. In contrast, the hippocampus contains mainly membrane ERα and few ERα nuclei [9]. In both regions, ERα can affect numerous neuronal plasticity processes including the expression of other ERs such as ERβ [10,11,12]. In the hippocampal CA1 region, estrogens increase the dendritic spine density and the expression of synaptic proteins, as well as activate local signaling pathways within spines [11,13]. In particular, estrogens regulate the expression of the neurotrophin receptor TrkB within dendritic spines and terminals in CA1 and the dentate gyrus (DG) [14], and TrkB is important for long-term potentiation and spine structure modification [15]. Estrogens also affect the mossy fiber pathway in the DG and CA3 in rodents, e.g., by inducing sprouting and elevating brain derived neurotrophic factor [16], as well as the opioid peptide dynorphin (Dyn) [17,18]. However, the role of nuclear and membrane ERα in these processes is poorly understood.

The design of mice that express either nuclear or membrane ERα has made it possible to begin to elucidate the tissue-specific roles of these receptors [5,19]. Thus, the aim of this study was to examine the effects of selective expression of membrane-only ERα (MOER) or nuclear-only ERα (NOER) on the distributions of proteins known to be involved in PVN function and estrogen-mediated plasticity in the hippocampus. We found that, in the PVN, the absence of nuclear ERα in MOER female mice resulted in increased nuclear ERβ. In the hippocampus CA1, which has robust membrane ERα expression, extranuclear ERα and pTrkB are decreased in both MOER and NOER female mice. In the dentate gyrus, pTrkB is increased in MOER mice but decreased in NOER female mice. In the CA3, only the NOER female mice showed a decrease in sprouting in mossy fibers. These studies demonstrate that disruption of estrogen signaling through either nuclear or membrane ERα can alter the expression of estrogen receptors and plasticity markers in the mouse hypothalamus and hippocampus.

## 2. Materials and Methods

### 2.1. Animals

Experimental procedures were approved by the Institutional Animal Care and Use Committees of Weill Cornell Medicine and Rockefeller University were in accordance with the 2011 Eighth Edition of the National Institutes of Health Guide for the Care and Use of Laboratory Animals. A total of 34 adult mice (24 females and 10 males) were used from three strains: (1) wildtype C57BL/6J mice; (2) knock-in (KI) MOER mice; (3) KI NOER mice. Briefly, a targeting vector was created by the UC-Davis Mouse Biology program via bacterial artificial chromosome recombineering that contained the cysteine-451 to alanine (C451A) mutation in ERα and inserted into the *esr1* locus in embryonic stem (ES) cells. After cell colony selection and confirmation by Southern blots with probes specific to the 5′ and 3′ end of the vector, ES cells were injected into blastocysts from C57BL/6NTac mice creating chimeric male mice. The neomycin-resistant cassette (neo) germline was confirmed by breeding the chimeric males with C57BL/6NTac females and assaying the offspring. Chimeric males were then crossed with Flpe-expressing females (stock #005703, Jackson Laboratories, Bar Harbor, ME, USA) to delete the neo cassette. Genotyping by PCR was performed using primers that specifically yielded a 400 bp gel band corresponding to C451A ERα with the neo gene excised [20]. WT mice were not littermates; however, they were age-matched. Mice were shipped from the Ellis Lab at UCI to Rockefeller University, housed 2–4 mice per cage with ad libitum access to food and water, and maintained on a 12 h light/dark cycle. Mice were allowed to acclimate to the vivarium for ~1 week prior to euthanasia. All female mice in this study experienced vaginal smear cytology; in wildtype mice, this was used to determine the estrous cycle stage. Wildtype mice were in proestrus on the day of euthanasia to parallel the MOER and NOER mice, which exhibited high estrogen levels [20,21]. 

Only females were used for the ERα and pTrkb hippocampal electron microscope studies, as our prior studies showed the most prominent effects on these markers were in females [9,14]. Both females and males were used to study ERβ labeling in the PVN since this marker is prominent in both females and males [8].

### 2.2. Tissue Preparation 

Tissue was prepared using previously described procedures [22]. Mice were deeply anesthetized with sodium pentobarbital (150 mg/kg, i.p.) and fixed by aortic arch perfusion, with 3–5 mL of saline (0.9%) containing 2% heparin followed by 30 mL of 3.75% acrolein and 2% paraformaldehyde (PFA) in 0.1 M phosphate buffer pH 7.4 (PB). Brains were extracted from the skull and post-fixed in 1.9% acrolein and 2% PFA in PB for 30 min at room temperature. To ensure consistency in brain comparisons, brains were blocked coronally between the cauda hippocampus and pons using a brain mold (Activational Systems). Brains were coronally sectioned (40 μm thick) on a VT1000X vibratome (Leica Microsystems, Buffalo Grove, IL, USA) and stored at −20 °C in cryoprotectant (30% sucrose, 30% ethylene glycol in PB) until immunocytochemical processing. 

### 2.3. Antibodies

**Dyn:** A polyclonal antiserum generated in guinea pig against Dyn (Peninsula Laboratories, Belmont, CA, USA) was used. The specificity of the Dyn antibody was shown in tissue sections showing the absence of immunolabeling when the antibody was adsorbed with the antigenic peptide [23].

**ER**α**:** A rabbit polyclonal antiserum (AS409) against the near-full-length peptide of the native rat ERα (amino acids 61 through the carboxyl terminus) was used (produced and generously supplied by S. Hayashi, Tokyo Metropolitan Institute of Neuroscience; Tokyo; Japan Cat# ER alpha, RRID: AB_2314382). The antibody recognizes the ERα occupied by ^3^H-estradiol [24,25]. The specificity of this ERα antibody has been demonstrated on immunoblots of uterine lysates from female rats which recognized one major band migrating at ~67 kD (the molecular weight of ERα) [26]. Additionally, immunoblots of ERα fusion protein and its degradation products demonstrated that the AS409 antibody recognized minor bands migrating at ~110 kD (likely the ERα/fusion protein complex), as well as major and minor bands migrating at ~67 kD and ~41–45 kD respectively (the degradation products of ERα, resulting from the purification process of ERα from the fusion protein). In pre-adsorption control trials of this antibody with purified ERα, no bands were detected on immunoblots, and immunolabeling was abolished in dendritic spines and presynaptic profiles in the CA1 of rat hippocampus, as visualized by electron microscopy [26]. 

**ERβ:** A rabbit antibody to ERβ (Z8P (discontinued) Zymed Laboratories, San Francisco, CA) was used. The specificity of this antibody was demonstrated previously by (1) absence of labeling in brain sections from ERβ knockout mice, (2) pre-absorption control with the antigenic peptide, and (3) dual labeling with mRNA using in situ hybridization [27]. 

**pTrkB:** A polyclonal rabbit antiserum against the activated form TrkB was used. This antibody was generated using a synthetic peptide (LQNLAKASPVpYLDIC) containing phosphorylated tyrosine 816 of rat TrkB. Specificity for this antibody was demonstrated by Western blot recognizing the activated full-length TrkB in 292 cells overexpressing TrkB receptors in hippocampal cells [28]. Additionally, pTrkB immunoreactivity (ir) is markedly decreased in olfactory bulb tissue and brains of TrkB receptor haploinsufficient mice [28]. The antibody does not recognize the truncated TrkB isoform [29,30,31,32,33]. Furthermore, the antibody does not recognize activated TrkA receptors in PC12 cells or other tyrosine-phosphorylated proteins [28]. This antibody has been used in prior electron microscopic studies in the mouse hippocampus [14].

**ZnT3:** A polyclonal rabbit antibody to ZnT3 (Cat # 197002; Lot # 197002; Synaptic Systems) was used. The specificity of this antibody has been demonstrated by Western blot (manufacturer’s data sheet). The pattern of ZnT3 labeling in mouse hippocampus with this antibody is identical to that observed in prior studies [34].

### 2.4. Light Microscopic Immunocytochemistry

**Immunolabeling procedures:** Free-floating sections were processed for immunoperoxidase labeling using the avidin–biotin complex (ABC) protocol [22]. To ensure identical labeling between groups, tissue sections were coded with hole punches in the cortex and pooled into containers to be processed together through all immunocytochemical procedures. Sections were first rinsed in PB to remove the cryoprotectant, and then incubated in 1% sodium borohydride in PB for 30 min to remove active aldehydes as tissues were fixed with acrolein. This was followed by 8–10 washes in PB until no gaseous bubbles remained.

Sections were sequentially incubated in (1) 0.5% bovine serum albumin (BSA) in tris-buffered saline (TS) for 30 min, (2) primary antisera (ERα, 1:8000 dilution; ERβ, 1:1000 dilution; Dyn, 1:10,000 dilution; ZnT3, 1:20,000 dilution) in 0.1% BSA and 0.25% Triton-X in TS for 1 day at room temperature (~23 °C), followed by either 1 day in the cold (∼4 °C) (Dyn, ZnT3) or 5 days in the cold (ERα, ERβ), (3) 1:400 dilution of donkey anti-rabbit IgG (Jackson ImmunoResearch Inc., West Grove, PA, USA) for 30 min, and (4) ABC solution at half the manufacturer’s recommended dilution (Vector Laboratories, Burlingame, CA, USA) in TS, 30 min. The bound peroxidase in the sections was visualized by a timed reaction (5 min ERα; 10 min ERβ; 4.5 min Dyn; 3.5 min ZnT3) in 3,3′-diaminobenzidine (DAB; Sigma-Aldrich Chemical Co., Milwaukee, MI, USA) and hydrogen peroxide in TS. Sections were washed in TS between each incubation. Sections were mounted on gelatin-coated slides, dehydrated through an ascending series of alcohols, and cover-slipped from xylene with DPX mounting medium (Sigma-Aldrich).

**Analysis:** Optical densitometric analysis on sections labeled for Dyn and ZnT was performed by an experimenter blinded to group identity. For this, light microscope photographs (10× magnification) from each region of interest (ROI) were taken on a Nikon Eclipse 80i microscope using a Micropublisher 5.0 digital camera (Q-imaging, Surrey, BC, Canada) and IP Lab software (Scanalytics IPLab, RRID:SCR_002775). The mean density value was obtained using ImageJ64 (ImageJ, RRID:SCR_003070) software for each ROI. For both Dyn and ZnT3, pixel density was determined from five ROIs: crest and central hilus of the DG and stratum lucidum (SLu) of CA3a, b, and c. To control for variations in overall illumination levels and background staining, the pixel density of an unlabeled tissue region (e.g., stratum radiatum of CA3) was subtracted from the ROI pixel density measurements. Prior studies verified the accuracy of this method through a strong linear correlation between average pixel density and neutral density values of gelatin filters with defined transmittances ranging from 1% to 80% [35,36]. The width of ZnT3 labeling in SLu in CA3a and b, as well as the length of ZnT3 labeling in stratum oriens (SO) between CA3b and CA3c, was measured using Image J64. For consistency, the notch where SO meets the molecular layer of the DG was used as the starting point of the length measure. 

In the PVN, ERα- and ERβ-labeled nuclei appeared as dense ovals, whereas extranuclear ERα and ERβ labeling was less distinct and filled the cytoplasm. ERα- and ERβ-labeled nuclei were counted manually, and the area of the PVN was calculated using Image J similar to prior studies [12]. The density of ERα or ERβ nuclei was calculated as the number of labeled cells/mm^2^. For ERα counts, the rostral level (approximately −0.9 mm from the bregma) of the PVN was assessed; for ERβ counts, the caudal level (approximately 1.20 mm from the bregma) was assessed.

### 2.5. Electron Microscopic Immunocytochemistry

**Immunolabeling procedures:** Free-floating sections were processed for immunoperoxidase as described above for light microscopy except that Triton-X was omitted from the primary antibody diluent. The antiserum to ERα was used at a 1:10,000 dilution and the antiserum to pTrkB was used at a 1:1000 dilution.

Hippocampal sections were processed for electron microscopy using previously described protocols [22]. Sections were post-fixed in 2% osmium tetroxide in PB for 1 h, followed by a dehydration series of alcohols and propylene oxide, and then embedded in EMBed 812 (Electron Microscopy Sciences (EMS), Hatfield, PA, USA) between two sheets of Aclar plastic. Ultrathin sections (70 nm thick) through the ROI were cut with a diamond knife (EMS) on a UCT ultratome (Leica). Sections were collected on 400 mesh thin-bar copper grids and counterstained with uranyl acetate and Reynold’s lead citrate. 

**Analysis:** All analysis was performed by experimenters who were blind to experimental groups. For quantitative studies, ultrathin sections from three mice in each experimental condition were analyzed at the tissue–plastic interface to minimize the difference in antibody penetration [22]. Photographs were taken on a Tecnai Biotwin (ERα) or CM10 (pTrkB) transmission electron microscope. For quantification of ERα profiles, all ERα-labeled profiles (e.g., dendrites, spines, axons, and terminals) were photographed at 20,000× in 2 grid squares (6050 μm^2^) per mouse in fields of the CA1 stratum radiatum between 50 and 150 μm below the pyramidal cell layer as in prior analysis [9]. pTrkB profiles were quantified from 20 random photographs at 13,500 magnification (220.5 μm^2^) from the stratum radiatum of CA1 and the hilus of the DG, which were taken from each mouse as in prior analysis [14].

Immunolabeled profiles were classified using defined morphological criteria [37]. Briefly, dendritic profiles contained regular microtubule arrays and were usually postsynaptic to axon terminal profiles. Dendritic spines were usually smaller than 0.2 μm in diameter, contacted by terminals forming symmetric synapses, and sometimes were contiguous with dendritic shafts. Unmyelinated axons were profiles smaller than 0.2 μm in diameter, contained a few small synaptic vesicles, and lacked a synaptic junction in the plane of the section. Axon terminal profiles had numerous small synaptic vesicles and had a cross-sectional diameter greater than 0.2 μm. Glial profiles contained no microtubules and conformed to the shape of surrounding structures. 

### 2.6. Figure Preparation

For light microscope photomicrographs, adjustments to brightness, sharpness, and contrast were made in Microsoft PowerPoint 2010. For electron micrographs, images were first placed in Adobe Photoshop 9.0 where resolution was increased to 300 dpi, and then images were adjusted for levels and sharpness using unsharp mask. Electron micrographs then were imported into Microsoft PowerPoint 2010, for additional changes to brightness, contrast, and sharpness so as to achieve uniformity in the appearance. Image adjustments were made to the entire image without alterations to labeling. Graphs were generated using Prism 8 software. 

***Statistical analysis:*** Data are expressed as means ± SEM. Significance was set to α < 0.05. Statistical analysis was conducted on JMP 12 software. A Student’s *t*-test was used for two-group comparisons. One-way analysis of variance (ANOVA) was used for three-group comparisons. Two-way analysis was used for comparing ERβ labeling across WT, MOER, and NOER males and females.

## 3. Results

### 3.1. Examination of Nuclear ERα Labeling in the Hypothalamus

ERα-labeled nuclei are prominent in the mouse PVN [8]. Thus, nuclear ERα labeling was examined at the light microscopic level in the PVN from WT, MOER, and NOER female mice (*n* = 4/group). Similar to prior studies [8], numerous ERα-labeled nuclei were detected in the PVN of WT mice, especially in the dorsal parvocellular division (Figure 1A,D). However, ERα-labeled nuclei were absent from the PVN of MOER mice (Figure 1B,E) while NOER mice showed similar nuclear labeling to WT mice (Figure 1C,F). One-way ANOVA, Welch-corrected, showed a significant genotype effect (F (2, 5.9) = 14.79, *p* = 0.0048). A post hoc Student’s *t*-test confirmed significant reduction in ERα-labeled nuclei in MOER compared to WT (0 ± 0 vs. 0.557 ± 0.098; *p* = 0.0109) and in MOER compared to NOER animals (0 ± 0 vs. 0.483 ± 0.094; *p* = 0.014). WT and NOER mice showed no significant differences in nuclear labeling (0.557 ± 0.098 vs. 0.483 ± 0.094; *p* = 0.60). 

### 3.2. Examination of ERβ Labeling in the Hypothalamus 

Both membrane and nuclear estrogen receptors come from the same mRNA [3,38]. Furthermore, ERα and ERβ are known to have some form of crosstalk with metabotropic glutamate receptors [39]. Thus, we examined nuclear ERβ-labeling in WT, MOER, and NOER female and male mice (*n* = 3–5 per genotype). Consistent with our prior study [8], ERβ-labeled nuclei were dispersed throughout the PVN (Figure 2A). Two-way ANOVA between female and male mice of the different genotypes showed a genotype effect (F(2160) = 3.723; *p* = 0.0470). A post hoc Tukey test showed that female MOER animals had a higher number of ERβ-labeled nuclei in the PVN compared to their WT counterparts (*p* = 0.05; Figure 2B). This increase in ERβ nuclei was opposite to the decreased ERα labeling seen in MOER mice (see above), suggesting a compensation for reduced membrane ERα. 

### 3.3. Examination of Extranuclear ERα Labeling in the Hippocampus

Our prior electron microscopic studies revealed numerous profiles containing extranuclear ERα-labeling within the rodent hippocampus, especially the CA1 region [9,26]. Thus, we next examined the distribution of ERα-containing profiles in the stratum radiatum of CA1 (Figure 3) from WT, MOER, and NOER female mice using electron microscopy (*n* = 3/genotype). ERα-labeled profiles were identified using standard morphological criteria [37] and quantified as previously described [9]. 

Consistent with prior studies [9,26], patches of ERα-ir were detected in dendritic shafts and spines (Figure 4A) in unmyelinated axons and terminals (Figure 4B), as well as glia profiles. The types of profiles labeled with ERα and the subcellular distributions of ERα were the same in WT and MOER/NOER mice. However, one-way ANOVA showed a significant genotype effect (F(2,6) = 4.7978, *p* = 0.056) present in the numbers of ERα-labeled dendritic spines. A post hoc Student’s *t*-test revealed that NOER mice had significantly fewer (*p* = 0.024) ERα-labeled dendritic spines compared to WT animals (Figure 4C). Likewise, MOER mice tended to have fewer (*p* = 0.069) ERα-labeled dendritic spines compared to WT mice. Additionally, one-way ANOVA showed a genotype effect (F(2,6) = 5.7917, *p* = 0.03) in ERα-labeled terminals. A post hoc Student’s *t*-test showed that both NOER (*p* = 0.018) and MOER (*p* = 0.047) had significantly fewer ERα-labeled terminals than the WT animals (Figure 4C). There were no significant differences in the total number of ERα-labeled dendrites or axons across the three genotypes (Figure 4D). Thus, in CA1, both MOER and NOER mice had reduced ERα-labeled dendritic spines and terminals compared to WT mice.

### 3.4. Examination of pTrKb Labeling in the Hippocampus

Previous electron microscopic studies showed that pTrkB was widely expressed in the mouse hippocampus, especially CA1 and the DG, and that the number of pTrkB-labeled profiles was highest when estrogen levels were elevated [14]. Thus, using immune-electron microscopy, we investigated the distribution of pTrkB-labeled profiles in the stratum radiatum of CA1 and the hilus of the DG (Figure 5 and Figure 6) from WT, MOER, and NOER female mice (*n* = 3 per genotype). Consistent with our prior study [14], pTrkB-ir was detected in dendritic shafts (Figure 5A and Figure 6A) and spines (Figure 6B), in unmyelinated axons (Figure 5B and Figure 6D) and terminals (Figure 5C), and in glia in both the CA1 and the DG. The types of profiles labeled with pTrkB and the subcellular distributions of pTrkB were the same in WT and MOER/NOER mice (Figure 6). 

In the CA1, one-way ANOVA, showed a significant genotype effect on the number of pTrkB-labeled dendrites (F(2179) = 5.670; *p* = 0.0041) and axons (F(2179) = 10.1883; *p* < 0.0001). A post hoc Student’s *t*-test revealed that WT animals had significantly more pTrkB-labeled dendritic profiles compared to MOER (*p* = 0.0019) and NOER (*p* = 0.0096) mice (Figure 5C). Likewise, WT mice had significantly more pTrKB-labeled axons compared to MOER (*p* < 0.0001) and NOER (*p* = 0.0007) mice (Figure 5C). Furthermore, there was a significant genotype effect (F (2179) = 8.6673, *p* = 0.0003) on the overall number of pTrkB-labeled profiles. A post hoc Student’s *t*-test showed significantly more pTrkB-labeled profiles in WT mice than MOER (*p* = 0.0015) and NOER (*p* = 0.0001) mice (Figure 5D). 

In contrast to CA1, the number of pTrkB-labeled profiles mostly varied between WT and MOER mice in the DG. One-way ANOVA revealed a genotype effect in the dendritic spines (F(2176) = 2.9994; *p* = 0.0524), terminals (F(1176) = 9.4033; *p* = 0.0001), and axons (F (2, 176) = 5.4001; *p* = 0.0053). Post-hoc Student’s *t*-tests revealed that WT mice compared to MOER mice had significantly fewer pTrkB-labeled dendrites (*p* = 0.0300), spines (*p* = 0.0341), and terminals (*p* < 0.0001; Figure 6E). However, WT mice had significantly more pTrkB-labeled axons compared to NOER mice (*p* = 0.0012; Figure 6E). MOER compared to NOER mice also had significantly more pTrkB-labeled spines (*p* = 0.0369) and terminals (*p* = 0.006; Figure 6E). Although the types of profiles with pTrkB labeling differed among WT, MOER, and NOER mice, the overall total pTrkB-labeled profiles was not significantly different (Figure 6F). 

Thus, in CA1, both MOER and NOER mice had reduced pTrkB labeling in dendritic and axonal profiles, and this resulted in an overall decrease in pTrkB-labeled profiles. However, in the DG, only NOER mice had reduced axonal pTrkB labeling compared to WT mice. In contrast, MOER mice had more pTrkB-labeled dendrites, spines, and terminals compared to WT mice. 

### 3.5. Examination of Dyn and ZnT3 in the Mossy Fiber Pathway

Our prior studies showed that Dyn levels in the mossy fiber pathway were elevated in select subregions in females compared to males but unchanged in ovariectomized ERα and ERβ knockout female mice [18]. Thus, we next investigated if Dyn-ir levels in the mossy fiber pathway were altered in female and male MOER and NOER mice (*n* = 4–6 per genotype). Consistent with our previous studies [17,18], dense diffuse Dyn-ir was seen in the hilus of the DG and in the SLu of CA3a-c (Figure 7A). The density of Dyn-ir was not significantly different in any subregion of either female or male MOER and NOER mice compared to WT (Figure 7B,C).

As mossy fibers contain ZnT3 [40], and as estrogens alter the expression of ZnT3 mRNA and synaptic vesicle zinc in female mice [41], we next examined the distribution of ZnT3 in the mossy fiber pathway of MOER and NOER mice. Similar to prior studies [34], dense diffuse ZnT3-ir was detected in the hilus of the DG and in SLu of CA3 (Figure 8A). Moreover, a band of ZnT3-labeled processes was observed in the inner region of SO contiguous with the hilus (Figure 8A). One-way ANOVA in the female, but not male, mice showed a genotype effect (F (2, 13) = 4.254, *p* = 0.0379) on the length of ZnT3 in CA3. A post hoc Student’s *t*-test showed that the female NOER, but not the female MOER, mice had significantly shorter (*p* = 0.0122) contiguous ZnT3-labeled fibers between the hilus and SO of CA3 compared to WT mice (Figure 8B–E). There were no sex or genotype differences in the width of ZnT3-labeled fibers in CA3a or CA3b (Figure 8F). Thus, these results suggest that the absence of nuclear ERα in female MOER mice reduces the sprouting ZnT3-containing mossy fibers in SO of CA3.

## 4. Discussion 

This study builds on our previous work to further elucidate the effects of selective expression of nuclear or membrane estrogen receptors on estrogen-mediated plasticity and signaling in the PVN and hippocampus. Importantly, the absence of nuclear and membrane ERα expression appeared to only impact the females in these brain areas. In the PVN, the absence of nuclear ERα increased nuclear ERβ. Moreover, in the hippocampus CA1, the absence of either nuclear or membrane ERα decreased extranuclear ERα and pTrkB in synapses. In contrast, in the DG, the absence of nuclear ERα increased pTrkB in synapses, whereas the absence of membrane ERα decreased pTrkB in axons only. However, the absence of membrane only ERα decreased the sprouting in mossy fibers in CA3.

Prior studies in PVN analyzing the distributions of nuclear ERα and ERβ in male and female mice revealed sex differences in their topographic distributions [8]. In particular, males have more ERα nuclei in the rostral neuroendocrine regions, whereas females have more ERβ-containing neurons in the caudal autonomic regions [8]. Previous studies showed that the loss of ERα in the PVN had no effect on blood pressure, whereas the loss of ERβ increased blood pressure [12,42]. Moreover, our recent research in a perimenopausal mouse model suggested that ERβ is necessary for the neuroprotective effects on hypertension in the PVN [12]. Thus, the increased expression of ERβ in the MOER females suggests that the loss of nuclear ERα expression may offer further protection in female mice to hypertension susceptibility. 

In the CA1 region of the hippocampus, estrogens modulate numerous synaptic plasticity processes [7,11]. Within the CA1, in vitro and in vivo estrogens increase dendritic spine density and synaptic proteins [11,13]. These estrogen actions are modulated largely by membrane ERα as few nuclear ERα are detected in CA1 [9,43]. This study found that, when either membrane or nuclear estrogen receptors were introduced, extranuclear ERα labeling was significantly reduced in CA1. Concomitant with these changes, pTrkB was reduced on pre- and post-synaptic processes in both MOER and NOER female mice. These results suggest that both membrane and nuclear ERα are important during development and adulthood, including maintaining plasticity responses in the CA1 region of the hippocampus.

In contrast to the CA1 region, in the DG MOER, females had greater pTrkB labeling on pre- and post-synaptic profiles compared to NOER or WT, even though the overall numbers of labeled profiles were not significantly different. In the DG, pTrkB colocalizes with Dyn in the mossy fibers [14]. However, the present study found no change in Dyn levels in the mossy fiber pathway in either the DG or the CA3. Our prior studies also showed that mossy fiber terminals contain abundant extranuclear ERβ [17]. Nuclear and extranuclear ERβ proteins and ERα mRNA are also present in newly born cells in the dentate subgranular zone [44]. In the DG, females have increased neurogenesis compared to males, and estrogens promote neurogenesis [45,46]. However, the separate roles of membrane and nuclear ERα and ERβ are unknown. Similar to the PVN, loss of extranuclear ERα could increase ERβ sensitivity in the DG.

In the DG, nuclear ERα is present in mossy fibers, interneurons that modulate the granule cell output [43,47]. In the interneurons, without the membrane ERα, there is reduced activity in mossy fibers. In the CA3, mossy fiber axons contain ERβ and ERα, but axon terminals contain mostly ERβ [9], making ERβ the predominate receptor in the mossy fiber pathway. However, estrogens have little effect on Dyn regulation in the mouse CA3 [18]; thus, the current results with MOER and NOER mice are consistent with previous work.

Our studies found that the sprouting of mossy fibers, identified by ZnT3, was reduced in NOER compared to WT female mice. Mossy fiber sprouting has been shown to be associated with increased synaptic plasticity in both females and males [16]. Moreover, estrogens have been shown to increase mossy fiber sprouting, although the roles of ERα and ERβ in this process are not well understood [16]. However, the CA3 mossy fiber region also contains GPER1 [48], and there is also evidence that GPER1 is involved in synaptic plasticity in this region [49,50]. These findings, along with those in other hippocampal regions, suggest reduced plasticity in the NOER mice.

## 5. Conclusions

The diversity of estrogen receptor expression in the brain suggests numerous roles for estrogens throughout life. Our results and prior research suggest that both estrogen receptor types are required for synaptic plasticity. In the CA1, an area with primarily extranuclear ERα, and the DG, which contains nuclear and extranuclear ERα and ERβ, pTrkb changed with both membrane and nuclear receptor loss, but the direction of the change was region-specific. In the hippocampal DG and CA3 regions, where there is less ERα expression overall, loss of nuclear or membrane ERα appeared to have less impact. PVN has both membrane and nuclear ERα, and only loss of nuclear ERα impacted ERβ. The observed sex differences in the PVN and hippocampus following the deletion of either nuclear or membrane ERα may be due to differences in estrogen levels, as well as organizational (development) or activational (adult) effects. Altogether, this study, along with research on MOER and NOER mice in other organs [19], generates new questions about the functions of membrane and nuclear ERs in the development of the brain and actions in adulthood.

## Figures and Tables

**Figure 1 biology-12-00632-f001:**
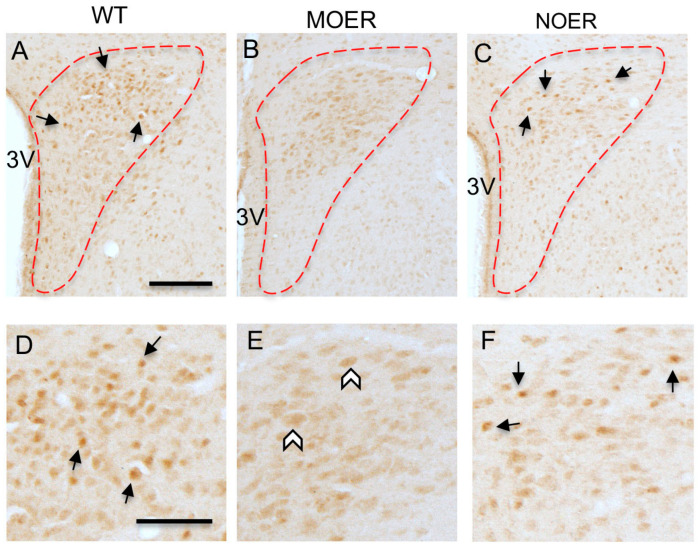
**Light microscopic distribution of nuclear ERα in the paraventricular nucleus of the hypothalamus (PVN) from WT, MOER, and NOER female mice.** Representative low-magnification micrographs of ERα labeling in the PVN (outline) from WT (**A**), MOER (**B**), and NOER (**C**) mice. Higher magnification of the dorsal parvocellular division showing numerous ERα-labeled nuclei (arrows) in the WT (**D**) and NOER (**F**) mice, but few ERα-labeled nuclei in the MOER mouse (**E**). Examples of extranuclear ERα labeling are also shown (chevrons); 3 V, third ventricle; scale bars (A) = 100 microns, (D) = 50 microns.

**Figure 2 biology-12-00632-f002:**
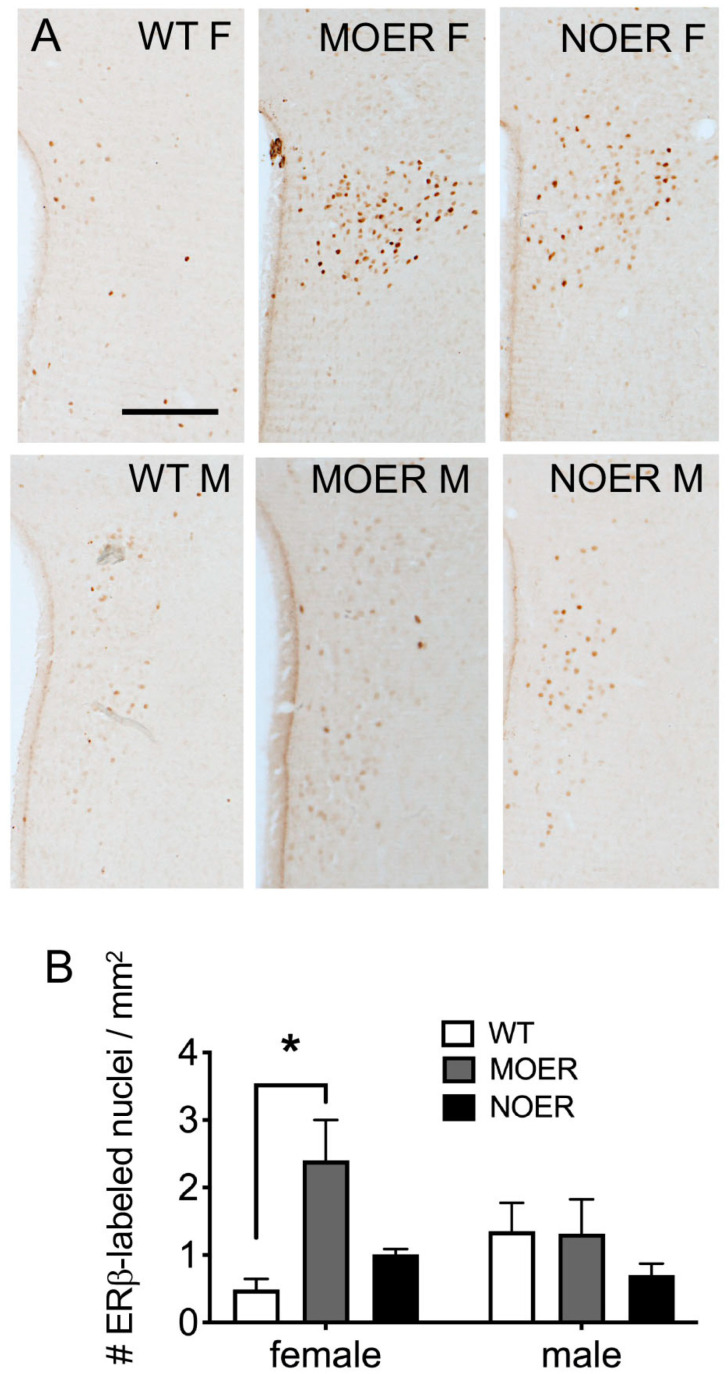
**Light microscopic distribution of nuclear ERβ in the PVN from WT, MOER, and NOER female and male mice.** (**A**) Representative micrographs of nuclear ERβ labeling in the PVN from WT, MOER, and NOER mice. (**B**) The density of ERβ-labeled nuclei (i.e., number of nuclei/mm^2^) in the PVN was greater in MOER females compared to WT females (*n* = 3 WT, 4 MOER, and 3 NOER females; *n* = 3 WT, 4 MOER, and 3 NOER males). * *p* < 0.05; scale bar (A) = 100 microns.

**Figure 3 biology-12-00632-f003:**
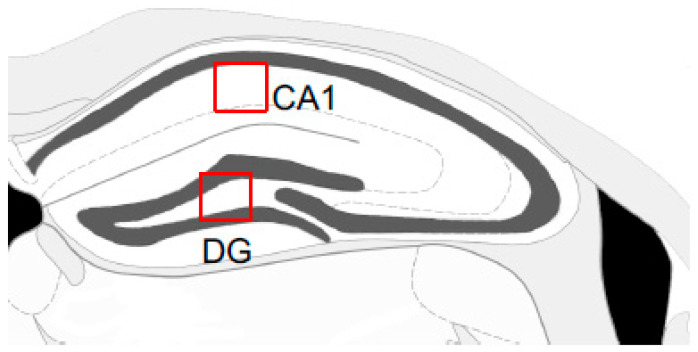
**Schematic of dorsal hippocampus showing regions sampled for EM**. The stratum radiatum of CA1 (top box) and the hilus of the dentate gyrus (DG; bottom box) were sampled for EM. Bregma level ~2.10 mm (modified from Allen Brain atlas https://atlas.brain-map.org (accessed on 4 December 2020)).

**Figure 4 biology-12-00632-f004:**
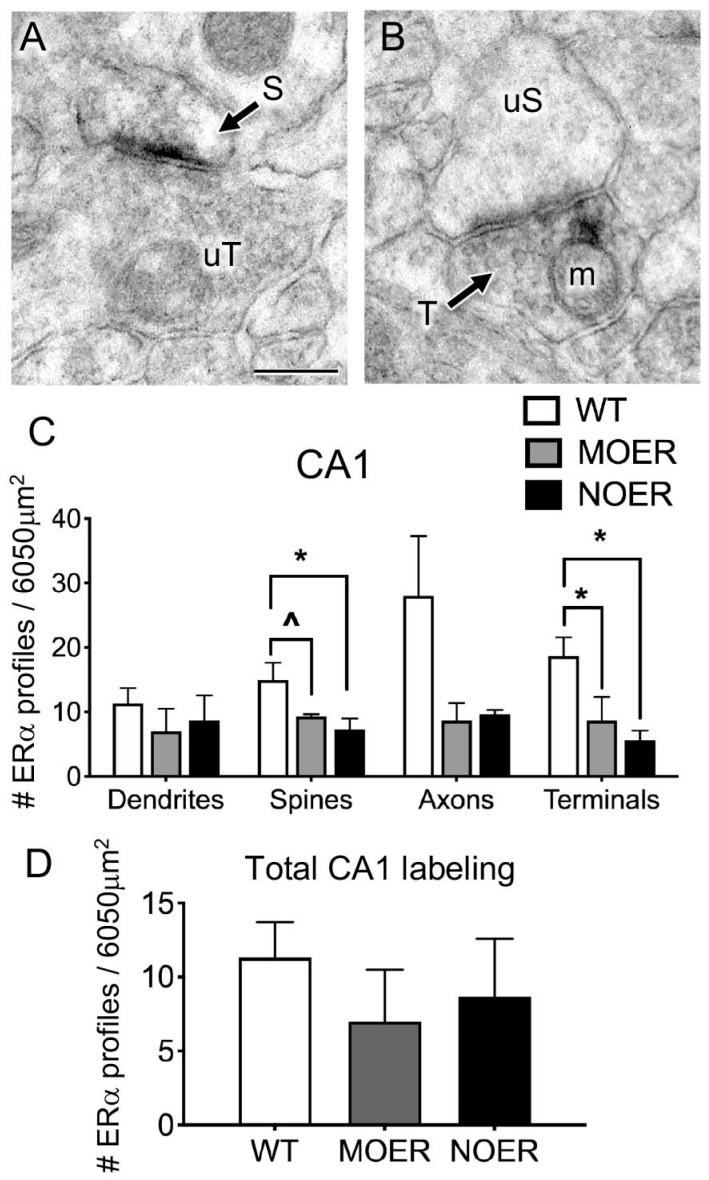
**Ultrastructural distribution of ERα labeled profiles in CA1 of WT, MOER, and NOER female mice.** (**A**) Example of an ERα-labeled dendritic spine (S) contacted by an unlabeled terminal (uT) from a WT mouse. (**B**) Example of an ERα-labeled axon terminal (T) synapsed on an unlabeled dendritic spine (uS) from a WT mouse. In this example, a cluster of ERα labeling is adjacent to a mitochondrion (m). (**C**) The numbers of ERα-labeled spines and terminals are reduced in MOER and NOER mice compared to WT mice. (**D**) Total ERα-labeled profiles were not significantly different among WT, MOER, and NOER mice (*n* = 3 mice per condition); * *p* < 0.05, ^ *p* = 0.069; scale bar = 250 nm.

**Figure 5 biology-12-00632-f005:**
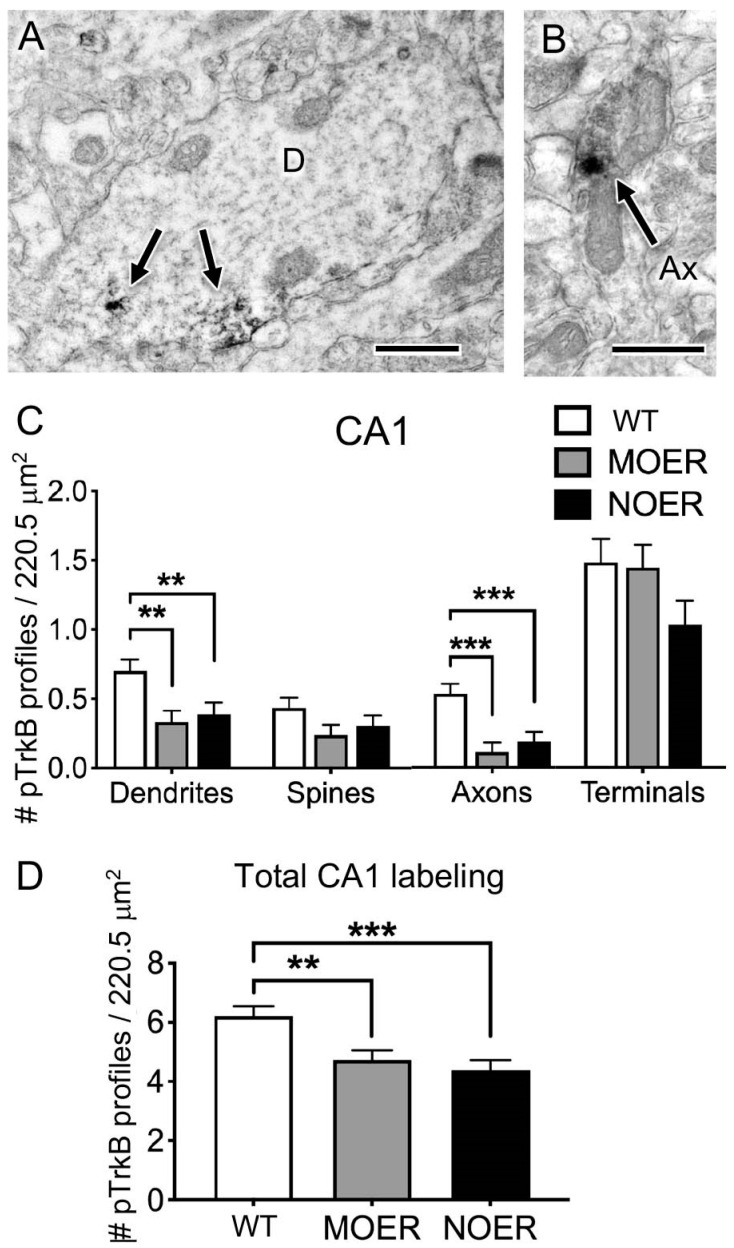
**Ultrastructural distribution of pTrkB labeled profiles in CA1 from WT, MOER, and NOER female mice.** (**A**,**B**) Representative micrographs showing pTrkB labeling in a dendrite (D) and axon (Ax) from WT mice. In the dendrite, the pTrkB immunoreaction product was affiliated with the plasma membrane (arrows). (**C**) The number of pTrkB-labeled dendrites and axons were significantly greater in WT animals compared to both MOER and NOER mice. (**D**) Total pTrkB-labeled profiles in CA1 were significantly greater in WT mice compared to MOER and NOER mice (*n* = 3 mice per condition); ** *p* < 0.005, *** *p* < 0.0001; scale bar = 500 nm.

**Figure 6 biology-12-00632-f006:**
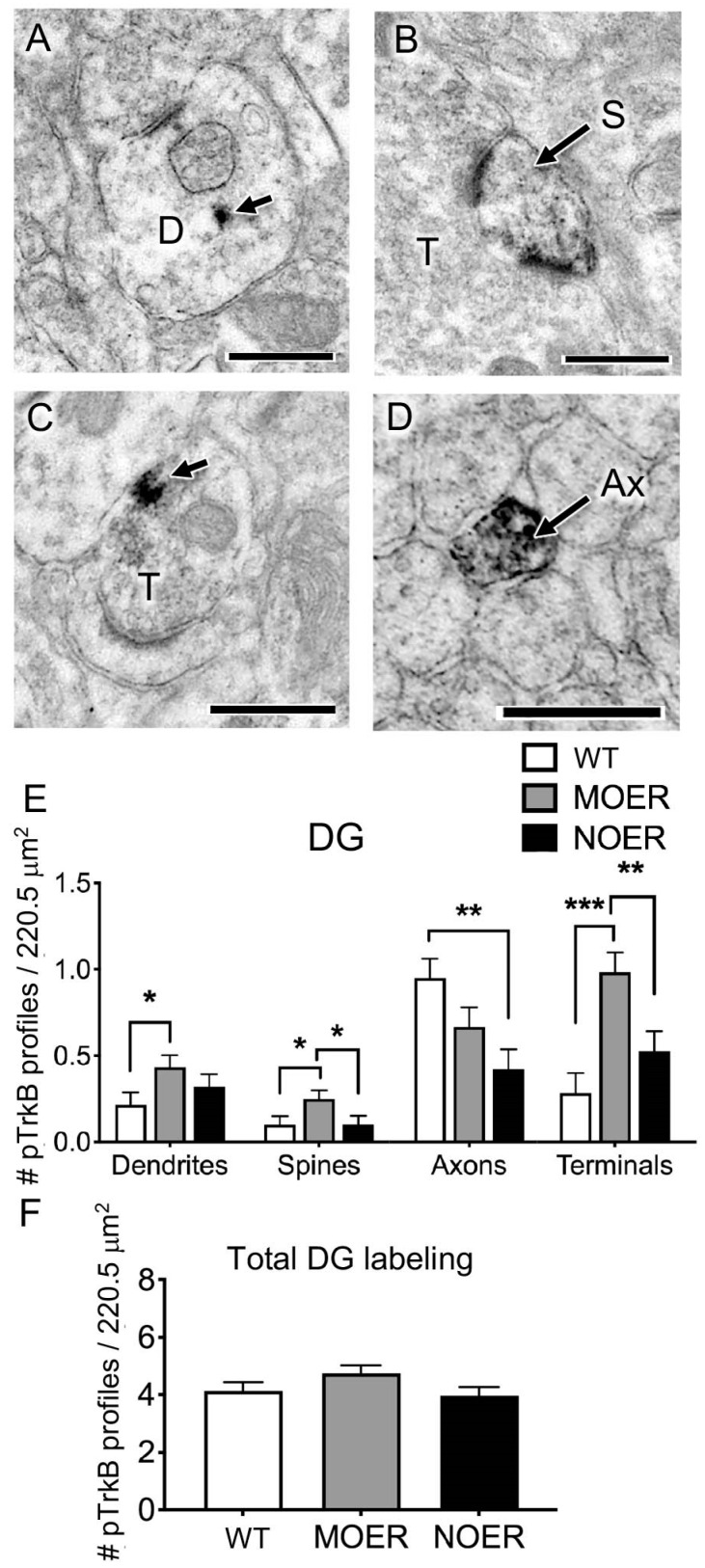
**Ultrastructural distribution of pTrkB-labeled profiles in the DG of WT, MOER, and NOER female mice.** (**A**–**D**) Examples of pTrkB labeling in a dendrite (D; **A**), dendritic spine (S; **B**), axon terminal (T; **C**), and axon (Ax; **D**) from MOER (**A**–**C**) and control (**D**). Punctate pTrkB labeling was found in the cytoplasm of the dendrite (arrow, **A**) and near the plasma membrane of the terminal (arrow, **C**). Photos from MOER (**A**–**C**) and WT (**D**) mice. (**E**). Compared to WT mice, MOER mice had significantly more pTrkB-labeled dendrites. Moreover, compared to both WT and NOER mice, MOER mice had more pTrkB-labeled dendritic spines and terminals. However, WT mice had significantly greater pTrkB-labeled axons than NOER mice. (**F**). Total pTrkB-labeled profiles in the DG were not significantly different among WT, MOER, and NOER mice (*n* = 3 mice per condition); * *p* < 0.05, ** *p* < 0.005, *** *p* < 0.0001; scale bar = 500 nm.

**Figure 7 biology-12-00632-f007:**
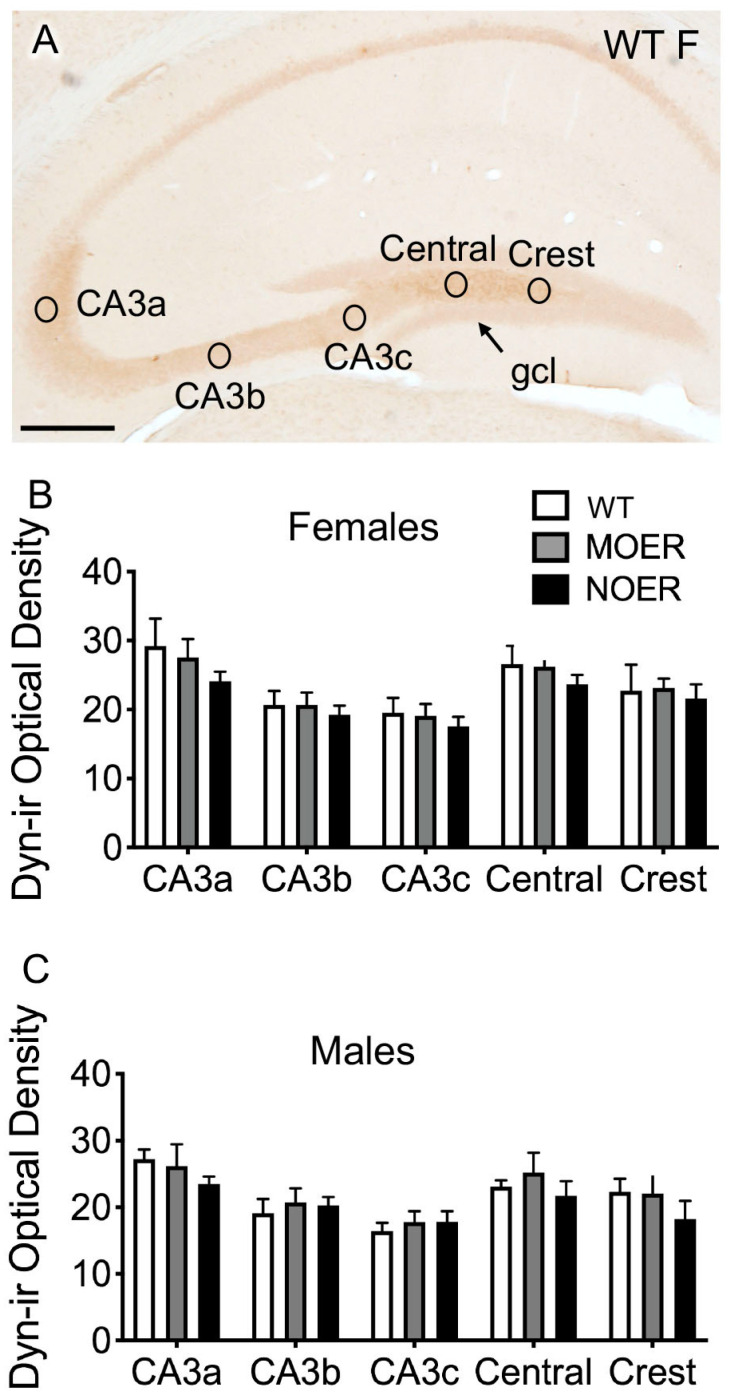
**Light microscopic localization of Dyn labeling in the mossy fiber pathway from WT, MOER, and NOER mice.** (**A**) Representative section of the dorsal hippocampus showing regions sampled (circles) for densitometry from the crest and central portion of the hilus of the DG and from the stratum lucidum (SLu) of CA3a, b, and c. (**B**,**C**) There were no significant differences between genotypes in the relative optical density of Dyn labeling in any region of the DG or CA3 from either female or male mice (*n* = 4 WT, 6 MOER, and 6 NOER females; *n* = 4 each for WT, MOER, and NOER males); scale bar = 100 microns.

**Figure 8 biology-12-00632-f008:**
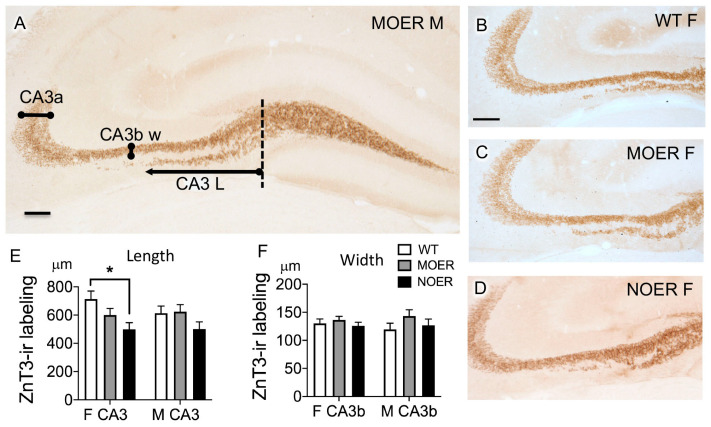
**Light microscopic localization of ZnT3 labeling in the mossy fiber pathway from WT, MOER, and NOER mice.** (**A**) Representative section of the dorsal hippocampus showing region of CA3 labeled for ZnT3 measured for width (CA3a and CA3b SLu) and length (region of CA3 SO contiguous with the hilus). For consistency, the notch where SO meets the molecular layer of the DG was used as the starting point of the length measure (vertical dashed line). (**B**–**D**) Examples show differences in the length of ZnT3 labeling in CA3 SO from WT (**B**), MOER (**C**), and NOER (**D**) female mice. (**E**) The length of ZnT3 labeling in CA3 SO was significantly shorter in the NOER compared to the WT female mice. (**F**) No significant differences were seen in the width of CA3b SLu from any of the mice from either sex (*n* = 4 WT, 6 MOER, and 6 NOER females; *n* = 4 each for WT, MOER, and NOER males); * *p* < 0.05; scale bar (**A**–**D**) = 100 microns.

## Data Availability

Data are available upon request. Please email Teresa A. Milner (tmilner@med.cornell.edu) or Elizabeth M. Waters (Elizabeth.waters@cooper.edu).

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
