# Peer review of "Both Nuclear and Membrane Estrogen Receptor Alpha Impact the Expression of Estrogen Receptors and Plasticity Markers in the Mouse Hypothalamus and Hippocampus"

_biology, 2023, doi:10.3390/biology12040632_

Round 1

Reviewer 1 Report

General Comments:

1.       The abbreviation of MOER and NOER seem to be confusing and inaccurate. MOER is not deficient in ERβ, isn’t it? If so, how about nERαKO mice or mERαKO mice?

2.       Do the authors use extranuclear ERα as a paraphrase of membrane ERα? I don't think this is accurate; extranuclear ERα also mean one that is localized in the cytoplasm. Or are they describing ERα expressed in axons and dendrites as extranuclear ERα? It would be easier to understand if they could unify their expressions.

3.       The wording of line 269 is confusing. Both membrane ERα and nuclear ERα are derived from the same mRNA, right? And ERβ is derived from a different transcript, isn’t it?

4.       It is easier to understand if a table summarizing the results is prepared, for example, using arrows to indicate increases or decreases in each item (ERα-labeled dendritic spines etc.).

5.       As the author also stated, the PVN is a long neuronal nucleus on the anterior-posterior axis and its function is known to be different. Therefore, I think it is necessary to include detailed information such as the distance from bregma of the sections used in the analysis and the number of sections. Also, a related question: are the levels of the section photographs shown in Figure 2 the same? Looking at the distribution pattern of ERβ, the NOER M picture seems to be at the rostral level.

6.       I don't think the quality of nuclear ERα staining in PVN is good. There are existing ERα antibodies, such as c1355 (Merk Millipore, Cat#06-935), which do not have such a high background, but why was the one used in this study chosen ? There are no needs to redo all the analyses, but I think it is important to compare and confirm the stained images with the current ones using other existing antibodies.

7.       The expression of ERα in reproduction-related brain regions is known to change with increasing or decreasing estrogen (Yamada et al., Neurosci Lett, 2009), but what about in the PVN and hippocampus? Because the changes in ERα- and pTrkB-labeled profiles in CA1 in this study are a phenomenon that was confirmed only in females, and the ovulatory cycle of the female mice used was not described, but I thought that it should be taken into account.

8.       Authors should mention in the Discussion the cause of the sex difference in this result.

9.      The authors have done mice deficient in membrane ERα and nuclear ERα genes in this experiment, but additional experiments with mice deficient in nuclear ERβ genes would be more revealing of the relationship between the three.

10.     The authors look at the effect of membrane ERα and nuclear ERα gene loss on morphological changes in this experiment, but do they think these changes also have functional and behavioral consequences (e.g., worse performance on simple spatial memory tests)?

11.     Do the authors think that the different effects of membrane ERα and nuclear ERα in different brain regions are function dependent? For example, more nuclear receptors are expressed in the PVN, which is a regulation of gene expression (slow temporal regulation), while the hippocampus requires faster regulation, etc.

12.     Overall, I think there are many careless mistakes as shown below. Although they are small, I think accurate descriptions are necessary because they affect the quality of the research itself.

Minor Comments

·        Alpha and beta have been written as a and b here and there. It should be corrected.

·        The reference figure for the scale bar of legend in Figure 1 is B. It should be D.

·        Which does the sentence in line 270 mean that "membrane ERα and nuclear ERβ" or "membrane ERα and membrane ERβ" ?

·        In Figure 2, the graph shows that WT M is the same as MOER M, but this does not appear to be the case in the photo. A representative photo should be selected that reflects the graph.

·        The title of Figure 2 is not accurate; either it is only for females, or there is also data for males. If the authors want to emphasize that the difference was found only in females, how about "MOER affected on the expression of ERβ in female mice" or something like that?

·        I am a little bothered by the opposite position of the axon and terminals bars in Figure 4 and Figure 5. It is a small thing, but it would be more comfortable if they were unified in one of them.

·        The text in line 333 should be changed to “The number of pTrkB-labeled dendrites and axons are significantly less in both the MOER and NOER mice compared to WT animals”. It sounds more natural to describe as “compared to WT”.

·        Please change the title of Figure 6 and the alphabet in the legend to be written in boldface to make it consistent with the other figures.

·        I don't think it's necessary to abbreviate as AEKO and BEKO in line 378, since there is no second appearance.

·        Figure 7 is of poor quality. The text is stretched horizontally, so the letters are stretched together.

·        The line 414 text also feels more natural with the opposite expression; not “The length of ZnT3 labeling in CA3 SO is significantly higher in the WT compared to the NOER female mice”, but “….is significantly shorter in the NOER compared to the WT female mice”.

Author Response

Reviewer 1

Point 1. The abbreviation of MOER and NOER seem to be confusing and inaccurate. MOER is not deficient in ERβ, isn’t it? If so, how about nERαKO mice or mERαKO mice?

Response 1.  The abbreviations MOER and NOER were used in the original report describing the mice. (Pedram et al. 2009, 2014). If membrane only or nuclear only, ERβ, mice become available, we expect that the names will be updated.

Point 2. Do the authors use extranuclear ERα as a paraphrase of membrane ERα? I don't think this is accurate; extranuclear ERα also mean one that is localized in the cytoplasm. Or are they describing ERα expressed in axons and dendrites as extranuclear ERα? It would be easier to understand if they could unify their expressions.

Response 2.  The term “extranuclear” is used to describe ERα-labeling outside the nucleus in the EM studies (e.g., dendrites, axons, terminals).  Extranuclear labeling would include cytoplasmic as well as membrane ERα. Indeed, our prior studies (e.g., Milner et al Endocrinology 149: 3306-3312, 2008) showed that these extranuclear profiles bind estrogens. 

Point 3. The wording of line 269 is confusing. Both membrane ERα and nuclear ERα are derived from the same mRNA, right? And ERβ is derived from a different transcript, isn’t it?

Response 3.  The sentence has been reworded (section 3.2, first sentence). 

Point 4. It is easier to understand if a table summarizing the results is prepared, for example, using arrows to indicate increases or decreases in each item (ERα-labeled dendritic spines etc.).

Response 4.  The graphical abstract, which was provided in the original submission, summarizes the results. Thus, a separate table is unnecessary.

Point 5. As the author also stated, the PVN is a long neuronal nucleus on the anterior-posterior axis and its function is known to be different. Therefore, I think it is necessary to include detailed information such as the distance from bregma of the sections used in the analysis and the number of sections. Also, a related question: are the levels of the section photographs shown in Figure 2 the same? Looking at the distribution pattern of ERβ, the NOER M picture seems to be at the rostral level.

Response 5.  The approximate bregma level from which the PVN sections have been taken have been added to the methods (section 2.4).  Although the distributions of the ERα and ERβ nuclei overlap in the PVN, our prior study (Contoreggi et al JCN 2021) showed ERα nuclei were prominent in the anterior to mid- levels of the PVN whereas ERb nuclei were more prominent in the caudal PVN.  Thus, the rostro-caudal level of the images in Figures 1 and 2 are not the same.

Point 6. I don't think the quality of nuclear ERα staining in PVN is good. There are existing ERα antibodies, such as c1355 (Merk Millipore, Cat#06-935), which do not have such a high background, but why was the one used in this study chosen ? There are no needs to redo all the analyses, but I think it is important to compare and confirm the stained images with the current ones using other existing antibodies.

Response 6.  This study utilized the ERα antibody from Hayashi (private source) since our prior studies show that it reliably and specifically identifies both nuclear and extranuclear ERα in acrolein/paraformaldehyde fixed tissue prepared for either light or electron microscopic immunocytochemistry (see Milner et al., JCN 429: 355-371, 2001).  Moreover, the use of this antibody allowed us to make direct comparisons with our prior studies.  We have previously used the Merk Millipore ERα antibody and found that it yields similar labeling to the Hayashi antibody in the hippocampus at the light and electron microscopic level (Milner, et al. Exp. Neurol. 212: 393-406. 2008). 

Point 7. The expression of ERα in reproduction-related brain regions is known to change with increasing or decreasing estrogen (Yamada et al., Neurosci Lett, 2009), but what about in the PVN and hippocampus? Because the changes in ERα- and pTrkB-labeled profiles in CA1 in this study are a phenomenon that was confirmed only in females, and the ovulatory cycle of the female mice used was not described, but I thought that it should be taken into account.

Response 7.  All female mice in this study experienced vaginal smear cytology; in wildtype mice it was used to determine the estrous cycle stage.  Wildtype mice were in proestrus on the day of euthanasia to parallel the MOER and NOER mice, which exhibit high estrogen levels.  This information has been added to the methods section (section 2.1). 

Point 8. Authors should mention in the Discussion the cause of the sex difference in this result.

Response 8.  The female mice used in this study were in proestrus.  Thus, we did not add a discussion of estrous cycle effect to the discussion.

Point 9. The authors have done mice deficient in membrane ERα and nuclear ERα genes in this experiment, but additional experiments with mice deficient in nuclear ERβ genes would be more revealing of the relationship between the three.

Response 9.  The study of mice deficient in nuclear ERβ would be interesting.  However, this is beyond the scope of this experiment which focused on ERα only.

Point 10.     The authors look at the effect of membrane ERα and nuclear ERα gene loss on morphological changes in this experiment, but do they think these changes also have functional and behavioral consequences (e.g., worse performance on simple spatial memory tests)?

Response 10.  Whether or not the expression of membrane only or nuclear only ERα has functional and/or behavioral effects is of interest.  Although beyond the scope of this study, it would be a topic of future studies.

Point 11.     Do the authors think that the different effects of membrane ERα and nuclear ERα in different brain regions are function dependent? For example, more nuclear receptors are expressed in the PVN, which is a regulation of gene expression (slow temporal regulation), while the hippocampus requires faster regulation, etc.

Response 11.  An interesting idea that we sincerely hope will be studied by our laboratory and others in the future.

Point 12.     Overall, I think there are many careless mistakes as shown below. Although they are small, I think accurate descriptions are necessary because they affect the quality of the research itself.

Response 12. We have corrected the items noted in the minor comments below.

Minor Comments

Point 13.  Alpha and beta have been written as a and b here and there. It should be corrected.

Response 13.  Changes in the alpha and beta symbols likely occurred when the manuscript was reformatted by the journal.  We have gone through the manuscript and converted the “a” and ”b” to alpha and beta.

Point 14.  The reference figure for the scale bar of legend in Figure 1 is B. It should be D.

Response 14.  The B has been changed to D in the Figure 1 legend.

Point 15.  Which does the sentence in line 270 mean that "membrane ERα and nuclear ERβ" or "membrane ERα and membrane ERβ" ?

Response 15.  This sentence has been rewritten (section 3.2).

Point 16.  In Figure 2, the graph shows that WT M is the same as MOER M, but this does not appear to be the case in the photo. A representative photo should be selected that reflects the graph.

Response 16.  We have replaced the MOER M ERβ photo with a photo that more accurately reflects the data.

Point 17.  The title of Figure 2 is not accurate; either it is only for females, or there is also data for males. If the authors want to emphasize that the difference was found only in females, how about "MOER affected on the expression of ERβ in female mice" or something like that?

Response 18.  The figure legend has been amended to include males.

Point 18.  I am a little bothered by the opposite position of the axon and terminals bars in Figure 4 and Figure 5. It is a small thing, but it would be more comfortable if they were unified in one of them.

Response 18.  The position of the axons and terminals on the graphs in figures 5 and 6 have been switched so that they match those in figure 4. 

Point 19.  The text in line 333 should be changed to “The number of pTrkB-labeled dendrites and axons are significantly less in both the MOER and NOER mice compared to WT animals”. It sounds more natural to describe as “compared to WT”.

Response 19.  This sentence has been changed as suggested (figure 5 legend).

Point 20.  Please change the title of Figure 6 and the alphabet in the legend to be written in boldface to make it consistent with the other figures.

Response 20.  Figure 6 has been changed as suggested.

Point 21. I don't think it's necessary to abbreviate as AEKO and BEKO in line 378, since there is no second appearance.

Response 21.  AEKO and BEKO have been deleted.

Point 22. Figure 7 is of poor quality. The text is stretched horizontally, so the letters are stretched together.

Response 22.  Figure 7 was not distorted in the submission that was uploaded.   We have reinserted the figure.

Point 23.  The line 414 text also feels more natural with the opposite expression; not “The length of ZnT3 labeling in CA3 SO is significantly higher in the WT compared to the NOER female mice”, but “….is significantly shorter in the NOER compared to the WT female mice”.

Response 23.  The wording has been changed as suggested. (Figure 8 legend)

Reviewer 2 Report

This is a follow-up study of the author’s previous works. Their study further grows a body of evidence surrounding differential roles of nuclear and membrane ERα on synaptic plasticity and will give a caution on the use of mice models expressing only nuclear or membrane ERα. Overall, the experiments are somewhat fragmented but the overall results support the author’s notion and will refine our understanding of the consequences of nuclear or membrane ERα disruption. However, the manuscript still has some concerns that should be addressed or clarified before publication.

Concerns:

I am confused about the origins of the mutant mouse models used. From line 96, some details of the development of NOER mouse were described but referred Pedram et al. (no.20) in the end. Is this a new mouse line different from Pedram et al.? If so, the targeting strategy and the validation study (Southern blot or long-range genomic PCR) should be provided. If not, it is enough to cite the original publications of the model development. The current description will cause confusion for readers. In addition, the origin of MOER mouse is not shown.

The breeding strategy to generate the study cohort is unclear. Were they littermates of MOER and NOER breeding? If they are not littermate or pool of WT from MOER/NOER intercrosses, the detail should be shown.

In Figure 1, how did you segregate ERα nuclear and extranuclear labeling? The criteria were not described in the Method section. It would also be greatly helpful if examples of extranuclear ERα IR are indicated in the Figure. In addition, is it right that the ERα-IRs in Figure 1B and C came from membrane/cytosolic ERα and nuclear/cytosolic ERα, respectively? In fact, some IRs are quite similar throughout the genotypes.

Lines 249-259: the units of the measurement are unclear. Are they the number of cells or the optical density per unit area or PVN area? In fact, which parameter was compared for ERα-IR is not mentioned in the Method section. 

Line 273: what was quantified and analyzed in two-way ANOVA?

Figure 2B: is this the quantification of the optical density of ERß-IR or the number of ERß cells per entire PVN area?

Figure 4: For the counting of ERα-IR profiling in EM, did you count a continuous ERα-IR in Figure 4A as 1, for example? And did you summate such profiles in larger EM images?

Line 305-308, Figure 4C/D, Figure 5 and 6: it is surprising that there were no significant differences in the total ERα profiles between genotypes despite clear genetic effects seen in Figure 4C. Are there other areas of ERα labeled profiles not shown in Figure 4C but considered in Figure 4D? The same question is also applied to Figure 5 and 6.

Examination of pTrkB labeling in the hippocampus (Lines 319-374): You mentioned that the number of pTrkB could be changed along with estrus cycle. Did you evaluate the estrus stages of the samples in this analysis? If so, tissue sampling/preparation along with estrus cycle evaluation should be described in the Method section.

Examination of ZnT3 in the mossy fiber pathway (Lines 393-405), Figure 8: small fluctuation of the cutting plane and section thickness during sample preparation would easily mess these evaluations of width/length of the measurements. Did you have any way to guarantee this would not happen? Stereology is ideal but if it is not applicable, would it better to quantitate them as the ratio of width of the IR profile per entire CA3 layers? I understand this would be challenging though.

Changes in pTrkB profiles (Figure 5 and 6) are not discussed enough in the association with the re-distribution of ERα among hippocampal compartments shown in Figure 4. I believe that you analyzed ERα and pTrkB profiles separately among the dendrites, spines, axons, and terminals because the ERα in those compartments differentially influences local pTrkB expressions. As CA1 has both ERα and pTrkB profile quantification, the discussion with the association of ERα and pTrkB in this region will give further insight on the relationship between local ERα and BDNF/TrkB signaling. In addition, should you discuss that a change in synaptic plasticity in NOER/MOER could result from re-distribution of ERα secondarily induced by the lack of either counterpart, as shown in Figure 4, 5, and 6?

Figure 4, 5, 6, and 7: the representative images from WT are not enough. Representative images of all comparison groups should be provided.

Figure 7A: Isn’t this image distorted horizontally?

x-axis of all figures: It would improve the readability by showing the units of the measurements (i.e., optical density/mm2 or the no. of ERα+ cells/PVN).

Author Response

Reviewer 2

Point 1.  I am confused about the origins of the mutant mouse models used. From line 96, some details of the development of NOER mouse were described but referred Pedram et al. (no.20) in the end. Is this a new mouse line different from Pedram et al.? If so, the targeting strategy and the validation study (Southern blot or long-range genomic PCR) should be provided. If not, it is enough to cite the original publications of the model development. The current description will cause confusion for readers. In addition, the origin of MOER mouse is not shown.

Response 1.  Details of both the MOER (Pedram 2009) and NOER (Pedram 2014) were added. Both lines were created on a C57BL/6 background.  The generation for each is not known.

Point 2.  The breeding strategy to generate the study cohort is unclear. Were they littermates of MOER and NOER breeding? If they are not littermate or pool of WT from MOER/NOER intercrosses, the detail should be shown.

Response 2.  The wildtype mice were not litter mates; however, they were age-matched.  This information has been added to the methods (section 2.1).

Point 3.  In Figure 1, how did you segregate ERα nuclear and extranuclear labeling? The criteria were not described in the Method section. It would also be greatly helpful if examples of extranuclear ERα IR are indicated in the Figure. In addition, is it right that the ERα-IRs in Figure 1B and C came from membrane/cytosolic ERα and nuclear/cytosolic ERα, respectively? In fact, some IRs are quite similar throughout the genotypes.

Response 3.  Details have been added to the methods section regarding how nuclear and extranuclear labeling were identified.  In addition to the examples of nuclear ERα labeling already shown, examples of  extranuclear ERα labeling have been indicated in Figure 1.

Point 4.  Lines 249-259: the units of the measurement are unclear. Are they the number of cells or the optical density per unit area or PVN area? In fact, which parameter was compared for ERα-IR is not mentioned in the Method section. 

Response 4.  The units of measurement used for the PVN analysis have been clarified.  It is numbers of labeled nuclei per unit area (section 3.1).

Point 5.  Line 273: what was quantified and analyzed in two-way ANOVA?

Response 5.  Data sets analyzed by two-way ANOVA have been clarified (section 3.2). 

Point 6.  Figure 2B: is this the quantification of the optical density of ERß-IR or the number of ERß cells per entire PVN area?

Response 6.  In the PVN, ERα- and ERb-labeled nuclei were counted manually and the area of the PVN was calculated using Image J similar to prior studies.  This has been clarified in the methods.

Point 7.  Figure 4: For the counting of ERα-IR profiling in EM, did you count a continuous ERα-IR in Figure 4A as 1, for example? And did you summate such profiles in larger EM images?

Response 7. ERα profiles were counted from two grid squares (6050 microns square) for each mouse.  The methods for quantifying EM profiles have been clarified in the methods (section 2.5).

Point 8.  Line 305-308, Figure 4C/D, Figure 5 and 6: it is surprising that there were no significant differences in the total ERα profiles between genotypes despite clear genetic effects seen in Figure 4C. Are there other areas of ERα labeled profiles not shown in Figure 4C but considered in Figure 4D? The same question is also applied to Figure 5 and 6.

Response 8.  The lack of differences in the total ERα -labeled profiles is likely due to the large error bars in some of the groups.  

Point 9.  Examination of pTrkB labeling in the hippocampus (Lines 319-374): You mentioned that the number of pTrkB could be changed along with estrus cycle. Did you evaluate the estrus stages of the samples in this analysis? If so, tissue sampling/preparation along with estrus cycle evaluation should be described in the Method section.

Response 9. See response 7, reviewer 1.

Point 10.  Examination of ZnT3 in the mossy fiber pathway (Lines 393-405), Figure 8: small fluctuation of the cutting plane and section thickness during sample preparation would easily mess these evaluations of width/length of the measurements. Did you have any way to guarantee this would not happen? Stereology is ideal but if it is not applicable, would it better to quantitate them as the ratio of width of the IR profile per entire CA3 layers? I understand this would be challenging though.

Response 10.  To ensure consistency in brain comparisons in all experiments, brains were blocked coronally between the cauda hippocampus and pons using a brain mold.  This has been added to the methods section. 

Point 11.  Changes in pTrkB profiles (Figure 5 and 6) are not discussed enough in the association with the re-distribution of ERα among hippocampal compartments shown in Figure 4. I believe that you analyzed ERα and pTrkB profiles separately among the dendrites, spines, axons, and terminals because the ERα in those compartments differentially influences local pTrkB expressions. As CA1 has both ERα and pTrkB profile quantification, the discussion with the association of ERα and pTrkB in this region will give further insight on the relationship between local ERα and BDNF/TrkB signaling. In addition, should you discuss that a change in synaptic plasticity in NOER/MOER could result from re-distribution of ERα secondarily induced by the lack of either counterpart, as shown in Figure 4, 5, and 6?

Response 11.  As ERα and pTrkB labeling was assessed in separate EM sections, it is not possible to know if the labeling is found in the same or different profiles.  Our analysis strategy does not extend to include evidence of changes in synapse number. The results with pTrkB are tantalizing but we do not want to overinterpret the results.

Point 12.  Figure 4, 5, 6, and 7: the representative images from WT are not enough. Representative images of all comparison groups should be provided.

Response 12.  The micrographs from the WT are used to show examples of the types of labeling in the difference types of profiles (i.e., dendrites, terminals, axons).  As the labeling did not appear different in the MOER and NOER mice, the inclusion of additional micrographs is not necessary.

Point 13.  Figure 7A: Isn’t this image distorted horizontally?

Response 13.  As noted in Reviewer 1 Point 22 above, Figure 7 was not distorted in the submission that was uploaded.   We have reinserted the image.

Point 14.  x-axis of all figures: It would improve the readability by showing the units of the measurements (i.e., optical density/mm2 or the no. of ERα+ cells/PVN).

Response 14.  The units of measurements have been added to the x axis of the figures.

Round 2

Reviewer 1 Report

I believe the author has answered each of my questions; I must question the appropriateness of the answer to Point 8.

I suggested to the authors that they discuss sex differences, but they stated that they did not add a discussion of estrous cycle effect to the discussion.

What I would like to know is what the authors think is the cause of the observed changes in WT, NOER and MOER in females but not in males? Could one of the causes be the concentration of estrogen ? (if females used in this study were  in proestrus, their estrogen concentration is higher than males, right?)

Author Response

REVIEWER 1

Point 1.  I must question the appropriateness of the answer to Point 8. I suggested to the authors that they discuss sex differences, but they stated that they did not add a discussion of estrous cycle effect to the discussion.

What I would like to know is what the authors think is the cause of the observed changes in WT, NOER and MOER in females but not in males? Could one of the causes be the concentration of estrogen ? (if females used in this study were in proestrus, their estrogen concentration is higher than males, right?)

[Prior point 8:  Authors should mention in the Discussion the cause of the sex difference in this result.

Prior response 8:  The female mice used in this study were in proestrus.  Thus, we did not add a discussion of estrous cycle effect to the discussion.]

Response 1:  It is established that MOER NOER females and proestrus females have high circulating levels of estrogen levels.  In this study we are unable to discern whether the difference in female and males is due to differences in estrogen levels, organizational (development) or activational (adult) effects of deleting nuclear or membrane ERa.  We have added a sentence to the conclusions to indicate unanswered question.

Reviewer 2 Report

I think that authors understood my questions and answered respectfully. However, the manuscript still contains the following critical/minor concerns. 

Regarding Response 3, the annotation of the arrowheads in Figure 1E pointing the typical examples for extranuclear ERα staining is missing from the figure legend.

Line 41: "ERb" should be "ERß" to keep the consistency.

This is a critical concern. In  Response 12, authors mention "..As the labeling did not appear different in the MOER and NOER mice, the inclusion of additional micrographs is not necessary." However, the study actually observed significant genetic effects in some of the quantifications. I cannot understand why authors responded like this. Their response does not match with their observation/discussion in the manuscript. If authors change their conclusion, they must reflect this change to the manuscript. These are also important for the scientific transparency.

Regarding Response 14, the unit of the measurement is added only to Figure 2B for ERß nuclear staining. The units are still missing from Figures 4-8. 

Author Response

REVIEWER 2

Point 1:  Regarding Response 3, the annotation of the arrowheads in Figure 1E pointing the typical examples for extranuclear ERα staining is missing from the figure legend.

Response 1:  A sentence referring to the arrows indicating extranuclear labeling has been added to the Figure 1 legend.

Point 2:  Line 41: "ERb" should be "ERß" to keep the consistency.

Response 2:  TheERb" has been changed to "ERß” in the abstract (line 38).

Point 3: This is a critical concern. In Response 12, authors mention "..As the labeling did not appear different in the MOER and NOER mice, the inclusion of additional micrographs is not necessary." However, the study actually observed significant genetic effects in some of the quantifications. I cannot understand why authors responded like this. Their response does not match with their observation/discussion in the manuscript. If authors change their conclusion, they must reflect this change to the manuscript. These are also important for the scientific transparency.

[Prior point 12.  Figure 4, 5, 6, and 7: the representative images from WT are not enough. Representative images of all comparison groups should be provided.

Prior Response 12.  The micrographs from the WT are used to show examples of the types of labeling in the difference types of profiles (i.e., dendrites, terminals, axons).  As the labeling did not appear different in the MOER and NOER mice, the inclusion of additional micrographs is not necessary.]

Response 3: We apologize that our prior response to point 12 was not clear and thus apparently misunderstood by the review.  The types of profiles (i.e., dendrites, spines, axons and terminals) labeled for ERα and subcellular distribution of ERα and pTrkB immunoreaction product did not differ between the WT and MOER/NOER mice.  Only the numbers of profiles differed between WT and MOER/NOER mice.  Thus, the inclusion of additional micrographs from the MOER and NOER mice in Figures 4-7 would have no added value.  However, sentences clarifying that the types of profiles containing ERα and pTrkB immunoreaction product and the subcellular distributions of ERα and pTrkB in WT and MOER/NOER mice were not different have been added to the results (sections 3.3 and 3.4).

Point 4: Regarding Response 14, the unit of the measurement is added only to Figure 2B for ERß nuclear staining. The units are still missing from Figures 4-8.

Response 4:  The units have been added to figures 4, 5, 6 and 8.  The measurements in figure 7 are relative optical density and thus there are no units to add. We have clarified that relative optical density was measured in the legend for figure 7. 

Round 3

Reviewer 1 Report

Your responses have convinced me.

Author Response

Thank you

Reviewer 2 Report

The latest author's response clarified my concerns.

I understand and respect author's decision not to include the representative images of the NOER/MOER mice.

However, I would like to suggest to show them as supplemental materials for research transparency. It should not harm the study and does not have to be well-formatted.

Author Response

Point 1:  I understand and respect author's decision not to include the representative images of the NOER/MOER mice.  However, I would like to suggest to show them as supplemental materials for research transparency. It should not harm the study and does not have to be well-formatted.

Response 1:  We have added a supplemental figure 1 with an electron microgragh showing ERa-labeling in a dendritic spine and terminal from a MOER mouse. Figure 6 contained pTrkB electron micrographs from both the control and MOER mice.  This has been clarified in the figure 6 legend and in the results (blue highlight).